# Prehospital and Emergency Care in Adult Patients with Acute Traumatic Brain Injury

**DOI:** 10.3390/medsci7010012

**Published:** 2019-01-21

**Authors:** Iris Pélieu, Corey Kull, Bernhard Walder

**Affiliations:** Division of Anaesthesiology, University Hospitals of Geneva, 12011 Geneva, Switzerland; Corey.Kull@hcuge.ch (C.K.); Bernhard.Walder@hcuge.ch (B.W.)

**Keywords:** head injury, trauma system, elderly, mortality

## Abstract

Traumatic brain injury (TBI) is a major healthcare problem and a major burden to society. The identification of a TBI can be challenging in the prehospital setting, particularly in elderly patients with unobserved falls. Errors in triage on scene cannot be ruled out based on limited clinical diagnostics. Potential new mobile diagnostics may decrease these errors. Prehospital care includes decision-making in clinical pathways, means of transport, and the degree of prehospital treatment. Emergency care at hospital admission includes the definitive diagnosis of TBI with, or without extracranial lesions, and triage to the appropriate receiving structure for definitive care. Early risk factors for an unfavorable outcome includes the severity of TBI, pupil reaction and age. These three variables are core variables, included in most predictive models for TBI, to predict short-term mortality. Additional early risk factors of mortality after severe TBI are hypotension and hypothermia. The extent and duration of these two risk factors may be decreased with optimal prehospital and emergency care. Potential new avenues of treatment are the early use of drugs with the capacity to decrease bleeding, and brain edema after TBI. There are still many uncertainties in prehospital and emergency care for TBI patients related to the complexity of TBI patterns.

## 1. Introduction

Traumatic brain injury (TBI) is a major cause of death and disability, and was identified as a major healthcare problem that affects 10 million people per year worldwide, particularly males [1]. Trends of TBI incidence are increasing even in high-income countries and correspond to a silent epidemic [2]. In the European Union, 1,000,000 TBI patients are admitted to hospitals each year, resulting in 50,000 non-survivors and ≥10,000 severely handicapped survivors [1]. Young males, very often under the influence of alcohol (46%), are most concerned [3,4]. Mild traumatic TBI accounts for approximately 70–90% of all TBIs. The incidence of patients with severe TBI was 11/100,000 persons/year, with a short term death rate of 30% in a study performed in Switzerland [5]. Similar mortality rates were observed in other high-income countries, for instance in Australia, with a death rate of 24.5% [6]. Mortality rates are significantly increased in elderly patients >65 years, whereby a mortality rate of 41% was reported [5]. Worldwide 90% of deaths, due to TBI occur in low or middle-income countries [7].

In a recent study, two main trauma mechanisms were observed: Falls (52.6%) and road traffic accidents (31.6%). The majority of elderly patients exposed to trauma suffer low-energy impact (i.e., falls <2 m from low heights). In the last years, TBI trauma mechanisms shifted from road traffic accidents and outdoor trauma, to indoor trauma and falls [5]. These changes in trauma mechanisms leading to TBI are associated with demographic changes in the general population, with more elderly people in industrialized countries [8].

The cognitive, mental, and physical impairment in survivors after TBI may result in a major burden for families and societies. This long-term morbidity of TBI is associated with a huge financial burden. In New Zealand, with 4.4 million inhabitants, the median cost per case for moderate or severe TBI was 36,648 $, and the lifetime cost of all TBI survivors was 147 million $ [9].

The aim of this narrative review was to summarize the limited evidence supporting the practice of prehospital and early emergency care at hospital admission after acute TBI. This narrative review is written by experts with a large scientific and clinical background in prehospital care in Europe. This review can support a critical appraisal of local prehospital care, in particular, prehospital care of elderly patients with acute TBI.

## 2. Acute Traumatic Brain Injury

### 2.1. Definition of Acute Traumatic Brain Injury

Traumatic brain injury (TBI), a form of acquired brain injury, occurs when a sudden trauma causes damage to the brain. The pattern of this acute cerebral injury is highly heterogeneous, but most often includes regional or global brain edema, and bleeding. Both alterations are space occupying lesions inside a minimally compliant cranium, which may contribute to intracranial hypertension and potential cerebral death. Apart from the primary brain lesion, a secondary brain injury related to inflammation and biochemical changes can be observed in most patients with severe TBI. This secondary injury begins within minutes of the primary injury and may last for several days, contributing to the final outcome.

The severity of TBI is assessed with a physiological estimation of consciousness [10,11], an anatomical estimation using Computed Tomography (CT), and a clinical assessment [12]. Severe TBI is defined physiologically as a total Glasgow Coma scale (GCS) < 9, or a subscale motor score (M-GCS) < 5, or anatomically as an Abbreviated Injury Score of head region (HAIS) > 3. Mild TBI is defined as a total GCS > 12 or as HAIS ≤ 1. Moderate TBI is bounded by the aforementioned numerical criteria.

The correlation between the GCS and severity of TBI has become less evident in recent studies [13]. The level of consciousness measured with the GCS is easy to identify if the clinical presentation is extreme, but the inter-rater variability is large for intermediate scores [14]. Alcohol or cannabis consumption before injury, and cognitive limitations (such as in elderly patients), may further decrease the accuracy of assessment. The motor component of the GCS (M-GCS) seems to be a more robust early risk factor. In the prehospital setting, an alternative score to the GCS can be the National Advisory Committee for Aeronautics severity score (NACA-SS) for the prediction of early mortality. In a Swiss prospective cohort of 677 patients with severe TBI, the NACA based model (including pupil reaction on scene, age), and a score based on M-GCS (including subscale motor score of GCS, pupil reaction on scene and age) were found to be equally accurate for the prediction of 14-day mortality [15].

### 2.2. Diagnosis of Traumatic Brain Injury

The diagnosis of TBI is often not evident immediately after the injury and differences with the final assessment in the hospital occurs [16]. The diagnostic evidence is increased if the event is witnessed, or if the mechanism of trauma is evident. However, unobserved low-energy and indoor trauma with decreased consciousness, but without coma may not be identified as TBI, particularly in elderly patients [17]. Furthermore, the differential diagnosis of a comatose state of unknown origin presenting in public is extensive [18]. Equally, medical problems precipitating a fall can incur a TBI.

Despite these difficulties in diagnosing TBI, a precise diagnosis and severity classification early after injury is essential for individualized medicine in the prehospital and emergency care setting, optimal pathway decision-making, and for an accurate prediction of outcome.

### 2.3. Risk Factors for Traumatic Brain Injury

In a Swedish cohort study, including 305,885 men conscripted for military service from 1989 to 1994, the independent risk factors found for mild TBI were low cognitive function, intoxication, and low socioeconomic status [19].

A long-term study observed that admissions of patients with moderate to severe TBI occurred more frequently on weekends, and during summer, with alcohol identified as a relevant risk factor [20].

### 2.4. Traumatic Brain Injury in Elderly

In high income countries, TBI is more prevalent in the elderly population (>65 years of age), as revealed in a cohort of severe TBI, within which 35% were elderly patients [5] (Table 1). Falls are the main trauma mechanisms in this population. [5,21,22]. The full degree of injury may take longer to manifest clinically in patients with preceding brain atrophy [23]. Patients >65 years of age showed a higher GCS at hospital admission compared with younger patients [24]. A recent observational study showed that for a given anatomical severity of TBI, older patients had a higher GCS than their younger counterparts [25].

These older patients often take drugs, including anticoagulants, and anti-platelet drugs. These drugs may aggravate the initial lesion, resulting in larger space-occupying intracranial bleeding. Recently, a meta-analysis revealed a correlation between anti-platelet therapy and post-traumatic cerebral hemorrhage (dds ratio (OR) 1.87; 95% confidence interval (CI) 1.27–2.74) [26]. Another study observed that a history of major bleeding, receiving concomitant therapy with aspirin and clopidogrel, presenting with headache or vomiting on hospital arrival, or abnormal GCS, confers an increased risk of intracranial bleeding following TBI [22]. Traumatic brain injury patients under concomitant anti-platelet therapy, are associated with higher mortality compared to those without [27]. For patients receiving anticoagulants, bleeding progression was shown to be prevented, and mortality reduced when the action of these drugs was quickly reversed [28]. In contrast to anticoagulants, no reversal therapy exists for anti-platelet agents.

In highly frail patients with a fall and TBI, ethical considerations should be addressed early to avoid futile care in the prehospital setting, and to implement palliative approaches in hospitalised patients, using community healthcare providers. Indeed, in a prospective observational cohort study of 161 elderly patients (≥60 years of age), with mild traumatic brain injury, frail patients had worse long-term outcomes, whereas age was not a significant predictor [29].

## 3. Prehospital and Emergency Care

### 3.1. Specificities of Prehospital and Emergency Care

Prehospital care, compared with acute hospital care, is characterized by limited diagnostic and therapeutic resources and an unfavorable environment for diagnostic or therapeutic interventions. Despite limited diagnostic possibilities, major clinical decision makings are performed: Hospital choice, choice of the means of transport, and choice of the immediate therapeutic interventions (see also Section 3.6). Therefore, and compared to hospital care, the risk of suboptimal care in the prehospital setting is increased, with potentially preventable deaths [31].

#### 3.1.1. Care Pathways

Roughly two pathways are available: A direct admission to a trauma center with neurosurgical facilities, or an admission to a hospital without immediate neurosurgery. Not every patient with mild TBI can go to a trauma center with neurosurgical facilities, as this would be an over-triage and would overcrowd such centers. On the contrary, every patient with a moderate or severe TBI should go directly to a trauma center with neurosurgical facilities, if not, it would be under-triage. In a study performed in the Netherlands, the accuracy of prehospital triage in selecting severely injured trauma patients was moderate; more than 20% of patients with severe injuries were not transported to an adequate trauma center [32].

Out-of-hospital emergency medical systems (EMS) should ensure the shortest delay possible between the sustainment of moderate or severe TBI, and the patient’s admission to a trauma center. A recent study showed that for patients with severe TBI, passing via a non-trauma center before transfer to a level 1 trauma center, in New-York, quadrupled the time (12.4 h ± 2.2 h vs. 3.1 h ± 1.2 h) to level 1 trauma center admission, resulting in a significant rise in mortality. However, this effect on mortality was only seen in patients with severe TBI. The overall mortality of patients with mild and moderate TBI did not differ if admitted directly or after transfer from a non-trauma center [33].

In urban regions, more often the closest non-trauma center rather than the closest trauma center was chosen, even in patients with major trauma [34]. This may have been done for initial patient stabilization and evaluation before transport to a trauma center.

#### 3.1.2. Means of Transport

There is controversy in how the risk of suboptimal prehospital care can be decreased; a hypothesis is that increased skills may decrease the risk of suboptimal management and may improve patient’s outcome [35]. Based on this hypothesis, in many European countries, physician-driven prehospital care was established, often with helicopter facilities and with more aggressive treatment possibilities, including orotracheal intubation. However, in many countries outside of Europe, including most regions of the North American countries, a paramedic-driven prehospital care system is favored.

There is no large generalizable randomized controlled trial to date that has tested staffing and means of transport in patients with TBI. However, there are some arguments that physician-driven prehospital care may be favorable for severe TBI patients.

In an underpowered randomized controlled trial performed in Australia, investing the impact of staffing and means of transport, the investigators observed a high non-compliance rate within the randomized groups and low evidence in favor of physician-driven prehospital care [36].

A large 2015 retrospective cohort study of the US National Trauma Bank observed that patients with TBI transported by helicopter to level 1 or 2 trauma centers, showed reduced mortality compared to ground transport teams. The absolute mortality risk reduction was 5.9% for patients admitted to a level 1 trauma center and 4.7% for patients admitted to a level 2 trauma center [37].

A retrospective study of 497 patients with severe TBI showed that physician led helicopter emergency services had a more favorable outcome and a lower mortality rate in comparison with paramedic crews, with similar times of arrival to a trauma center [38].

Another study, that included 334 patients with severe TBI admitted to level 1 trauma centers, and performed in the Netherlands, observed that a physician-based helicopter emergency medical service was more frequently deployed in patients with severe TBI in the presence of extracranial injuries. Despite higher injury severity levels, six-month mortality rates were equivocal [39].

Data on prehospital severe TBI management using physician-staffed, or paramedic-staffed systems, was retrospectively analyzed: Advanced airway management was performed in 16% of the patients in the paramedic-staffed EMS group, and in 98% of the patients in the physician-staffed-EMS group (*p* < 0.001). Mortality was significantly lower, and the neurological outcome was better in patients in the physician-staffed EMS group compared to the paramedic-staffed EMS group [40].

In contrast to these studies, a retrospective cohort-matched study of patients with isolated blunt severe TBI analyzed the effect of more systematic prehospital intubation and a physician on scene on outcomes. The study compared two different EMS, one from Bern, Switzerland, (physician-staffed EMS), and one from the United States (paramedic-staffed EMS). Patients in the Bern cohort had significantly longer scene times and more frequent prehospital endotracheal intubation when compared with the US cohort. In-hospital mortality was not significantly different between these cohorts [41].

Independent of skills and means of transport, there is a general agreement that prehospital care has to be integrated into an established and organized trauma system, which may substantially decrease the risk of suboptimal care [42].

### 3.2. Patient-Relevant Outcomes after Prehospital and Emergency Care

The outcome of prehospital and emergency care is focused on survival; non-survival patients after severe TBI die mostly within the first 3 days. Therefore, prehospital and emergency physicians consider short-term mortality as the most relevant outcome. Short-term mortality can be defined as in-hospital mortality, mortality at 14 days, or 30 days. The aim of prehospital and emergency care is to decrease early, preventable death after TBI.

Some authors have proposed to use the GCS at 14 days as a surrogate marker of neurological outcome in patients with severe TBI [43]. Recently, a Glasgow Outcome at Discharge Scale Score was proposed, which was performed at the initial hospital discharge [44]. After external validation, this new Glasgow Outcome Score could be a relevant outcome measure and could be of interest for healthcare providers in the prehospital and emergency care setting.

### 3.3. Early Risk Factors Associated with Outcome

The IMPACT database on TBI [45] consisted of eight randomized controlled trials and three observational series extending over the period 1984–1997. Overall, 9205 patients with moderate and severe injuries were included. A relevant association between the severity of TBI and outcome was observed. The higher the severity of brain injury (attested by GCS, pupil reactivity, and CT class), the higher the risk of poor outcome at 6 months [46]. Increasing age was also strongly associated with poorer outcome (OR 2.14; 95% CI 2.00–2.28) [47].

Arterial hypotension, (systolic blood pressure <90 mmHg), was an identified prehospital risk factor for mortality [48]. A study of major traumatic brain injury found that the depth, and duration of out-of-hospital hypotension were strongly associated with increased mortality [49]. Hypothermia (temperature <35 °C) was also identified as a prehospital risk factor for mortality [48,50]. Hypoxemia, (with a pulse oximeter oxygen saturation <90%), was a prehospital risk factor for either impaired consciousness in survivors at 14 days, or a lower GCS score at hospital discharge [48,51].

### 3.4. Early Prediction of Outcome

Accurate prognosis of TBI based on prediction models with high performance is required to counsel patients and proxies. Accurate prediction models can assist in the allocation of health care resources, allow stratification of patient groups, and be relevant in investigations testing the effectiveness of novel therapies. Different prediction models were developed in the last years, with most based on an estimation of TBI severity, pupil reaction, and age.

The IMPACT core model used the M-GCS together with pupillary reactivity and age to predict mortality [46].

In the pooled population, an unreactive pupil, or pupils, or lower motor scores, were significantly associated with an unfavorable outcome (OR 2.71 respectively OR 7.31). Furthermore, a significant change between prehospital and in-hospital admission M-GCS scores, as well as between in-hospital admission and study enrolment M-GCS scores were observed; a hint that consciousness is dynamic in the first hours after TBI [52]

The Abbreviated Injury Score (AIS) is a well-defined anatomical scoring system of trauma. The severity of each injury in seven body regions is graded on a scale from 1 to 6. In most cases, accurate grading requires diagnostic imaging. The AIS of the head region (HAIS) is a validated prognostic factor in TBI. The HAIS is principally based on initial CT imaging findings. A prediction model based on: HAIS, pupil reaction, and age, was not inferior when compared with a prediction based on the IMPACT core model [53]. This new model based on HAIS may be used in the case of an invalid or absent GCS.

An extended model with M-GCS and HAIS may improve outcome prediction. A large epidemiological cohort multi-center study compared the predictive performance of such an extended predictive model, including M-GSC, pupil reaction, age, HAIS, and the presence of multiple trauma, for short term mortality against a reference predictive model, including M-GCS, pupil reaction and age (IMPACT core model). This study, which included 808 severe adult TBI, observed that the extended predictive model had a statistically significant higher predictive performance [54].

### 3.5. Diagnostic Strategy

#### 3.5.1. Diagnostic Strategy in the Prehospital Setting

Apart from a general clinical check, vital signs are measured after the arrival of a preoperative care team on scene. Neurological evaluation includes the estimation of consciousness, pupil reaction, and neurological symmetry. Most often, the GCS is used for consciousness estimation, and for subsequent hospital pathway selection (Table 2). Decreased GCS is an indication for direct admission in a trauma center with permanent neurosurgical facilities.

Nakamura et al. observed that for field triage, the sensitivity of GCS progressively decreased with increasing age above 60, which risks suboptimal care [55]. Therefore, it is a challenge to avoid underestimation of TBI severity in elderly patients. Potentially, trauma systems need to develop new guidelines for the prehospital management of the elderly, with supplementary assessments apart from GCS. It is also of interest to observe that elderly patients have been treated less often by a certified emergency physician, compared to younger patients [5].

Independent of age, the risk of inadequate triage may also be related to the dynamic changes of TBI severity estimation with GCS. To decrease this risk, repeated vital signs and neurological evaluations should be performed at regular intervals, because some patients may develop a secondary neurological deterioration, [56] requiring a change of the intended clinical pathway.

#### 3.5.2. Diagnostic Strategy in the Emergency Department

Apart from the estimation of TBI severity, the search for other relevant lesions is of high importance in the emergency department. A large European prospective cohort of trauma patients found that patients with a GCS score <15 had a higher risk for cervical fractures/dislocations and spinal cord injury than patients with a normal GCS. Therefore, diagnostic investigations should include computed tomographic imaging of the cervical spine in patients with severe (GCS < 9) or moderate (GCS 9–13) TBI [57]. Furthermore, careful cervical spine protection before computed tomographic imaging may be indicated.

Rapid identification and control of hemorrhage in traumatic intracranial hemorrhage is important, since the majority of expansion occurs within the first 24 h after injury and hemorrhage expansion is an independent risk factor for mortality.

Concerning patients with mild TBI, large center variation exists in policies for diagnostics, admission, and discharge decisions in the emergency department. Guidelines are not always operational in all centers, and reported policies systematically diverge from what is recommended in those guidelines [58].

### 3.6. Immediate Therapeutic Interventions

The priority of prehospital and early emergency care for patients with TBI is cardiopulmonary stabilization and the prevention of secondary brain injury (Table 2) associated with a life-threatening rise in intracranial pressure.

#### 3.6.1. Respiratory Failure

After a TBI the function of respiratory or breathing control centers may be altered. Furthermore, dysphagia may be associated with TBI, which can be an indication for orotracheal intubation.

The potential benefit of prehospital intubation and positive pressure ventilation is controversial. In a prospective study, 352 adult trauma patients with severe TBI (with a Glasgow Coma Scale score of 3–8), were hand-matched to 704 controls. Paramedic rapid sequence intubation was associated with an increase in mortality compared with matched historical controls, caused mainly by hyperventilation [59]. In Finland, among unconscious patients with severe isolated TBI, 1-year mortality was significantly higher if they were managed by paramedic-staffed prehospital teams when compared to physician-staffed teams (57% vs. 42%, *p* = 0.001). On arrival to the emergency department, hypoxia was common in the patients treated by the paramedic-staffed EMS, while in the physician-staffed EMS, almost none of the patients were hypoxemic [40].

It is possible that a subgroup of patients suffering from severe TBI with hypoxemia on scene, or with long distances to a trauma center, may benefit from prehospital intubation, ventilation with strict normocarbia, and non-excessive administration of oxygen (spO_2_ between 92 to 95%) [60].

#### 3.6.2. Arterial Hypotension and Shock

Shock and arterial hypotension after TBI may be an indicator for supplementary organ lesions, and a search for supplementary lesions should be performed immediately, as about one third of TBI patients are patients with multiple trauma [5]. A meta-analysis observed a lower mortality in patients with traumatic hemorrhagic shock, receiving hypotensive resuscitation [61].

There is an ongoing debate about the optimal cut-off to treat hypotension in patients with TBI, and studies have proposed higher systolic blood pressures (SBPs) as thresholds for treatment [62,63]. In the latest version of the Brain Trauma Foundation guidelines, it was recommended to maintain SBP > 100 mmHg for patients aged 50–69 years, or >110 mmHg for patients aged 15–49 years, or >70 years. A recent retrospective cohort study from the Japan Trauma Data Bank, including 12,537 severe TBI patients, that were divided into two groups according to age, <60 years, and >60 years. In patients <60 years, SBP of 120–129 mmHg at hospital admission was associated with the lowest odds for mortality. In patients >60 years, SBP of 130–139 mmHg at hospital admission was associated with the lowest odds for mortality. They suggest that the definition of hypotension after TBI should be reconsidered [64].

In patients with traumatic brain injury, intravenous resuscitation fluids are fundamental components of the restoration and maintenance of systemic and cerebral circulations. There is uncertainty about the best choice of fluids, due to the lack of adequately powered randomized controlled trials. The exception is albumin: Higher mortality rates were observed among patients with severe traumatic brain injury (GCS score, 3 to 8) who received 4% albumin, compared to those who received saline. These findings suggest that saline is preferable to albumin during the acute resuscitation of severe TBI, at least in the ICU setting [65].

Trauma patient mortality increases with the magnitude of hemodynamic instability, and the extent of injury with bleeding risk. Prehospital blood product transfusion in trauma care remains controversial, due to poor-quality evidence, and costs. No survival advantage with prehospital red blood cell availability was recently observed [66]. Maintaining a hemoglobin concentration of approximately 10 g/dL was proposed some years ago to improve cerebral oxygenation in patients with traumatic brain injury and to reduce neurological injury, particularly during the acute recovery period when the brain is most vulnerable to ischemic insults. In a retrospective cohort study of patients with moderate or severe traumatic brain injury admitted to the intensive care unit, red blood cells transfusions were associated with an increased risk of death, neurological and traumatic complications, and a longer length of stay [67]. A randomized trial, which compared hemoglobin transfusion thresholds of 7 g/dL versus 10 g/dL did not observe a difference in favorable outcomes. However, more thromboembolic events were observed with the higher threshold [68]. Therefore, most centers today have a lower hemoglobin threshold for transfusion.

#### 3.6.3. Hypothermia after Traumatic Brain injury

Low body temperatures (<35 °C) in general trauma patients in the prehospital setting, is associated with poor outcomes, including mortality [69,70]. In victims of a major TBI, a significant association between abnormal temperature, (measured in the initial trauma center), and poor outcomes was observed [71]. Hypothermia is also related to uncontrolled bleeding in the prehospital and emergency care setting [72]. Therapeutic hypothermia in the intensive care unit has also been abandoned, based on results of a prospective randomized trial involving patients with traumatic brain injury and intracranial hypertension, where hypothermia plus standard care did not result in improved outcomes [73].

## 4. Uncertainties and Further Research

There are many uncertainties in prehospital and early emergency care of TBI patients, related to the complexity of TBI patterns. There is however, a certain consensus that hypotension, hypothermia, and hypoxia contribute to the aggravation of TBI. Ideal individualized treatment, with optimal cut-offs needs to be established in further research. The focus of research in prehospital and emergency care is on improved identification and stratification of TBI patients for adapted treatments and early interventions, decreasing intracranial bleeding and brain edema.

Improved triage in patients with TBI, to avoid the risk of suboptimal care in prehospital and emergency care requires improved early diagnostics. Research is focused on specific and rapidly available cerebral biomarkers and mobile imaging. Rapidly available biomarkers at hospital admission or point-of-care biomarkers in the prehospital period, with a high prediction of severity of TBI would change medical decision making. Many biomarkers have been extensively studied, however, none of these biomarkers is currently used in clinical practice. The focus is actually to identify patients with suspected TBI not needing a trauma center [74,75]. However, no biomarkers are capable of delivering results in the minutes following blood sampling. Prehospital CT imaging with mobile devices was proposed to improve correct triaging in patients with suspected TBI [76]. Further investigations are needed to estimate the feasibility of CT imaging in the everyday clinical practice of prehospital care without radiological specialists. Ocular ultrasonography is a non-invasive method for the detection of intracranial hypertension. With this method, the optic nerve sheath diameter can be measured [77]; the larger the diameter, the higher the intracranial pressure. This diagnostic method may be the base for very early goal-directed treatment of intracranial hypertension, which is a surrogate marker of unfavorable outcome [78].

A potentially relevant early intervention after TBI could be treated with tranexamic acid, which could reduce intracranial bleeding. Tranexamic acid was tested in the CRASH-3 trial [79,80]. Results are expected in the near future.

Many treatments were tested to decrease inflammation and brain edema after TBI. No treatments were found efficient. However, all these treatments were tested after hospital admission. Earlier administration of these treatments, when brain edema is beginning, should be investigated.

An actual weakness of prehospital and emergency care is the low emphasis on precipitating factors of TBI, which are potentially relevant for patient outcome. Such factors include, alcohol and drug consumption, social and psychological conditions. Such factors should be considered in protocols for prehospital and emergency care.

A research agenda for geriatric patients with TBI should be established for, prehospital and emergency care, in-hospital management, and rehabilitation. This research is urgently needed, due to the increase in geriatric patients, and should also integrate healthcare management concepts and ethical aspects to avoid futile treatments and unnecessary distress.

## 5. Conclusions

In the absence of highly efficient treatments, integrated healthcare management of TBI is absolutely essential for improved outcome. Skilled and highly trained healthcare providers over the whole chain of brain trauma care, including prehospital and emergency care, may also contribute to improved results. Given the complexity of TBI, only patient-relevant outcome assessments, such as mortality, disability and health-related quality of life, should be used for safety and research purposes.

## Figures and Tables

**Table 1 medsci-07-00012-t001:** Differences related to age after severe traumatic brain injury (TBI) in adult patients based on reference [30].

	Younger (≤65 Years Old)	Elderly (>65 Years Old)
**Initial observations**		
Distribution in %	65.3%	34.7%
Age peak per age class	20.0–29.9 years	60.0–69.9 years
Incidence per age class	7.9/100,000/year	22.4/100,000/year
Median GCS on scene	8	12
Median GCS in ED	3	8
In-hospital severity of TBI	HAIS 4-5	HAIS 4-5
Main drugs contributing to severity	none	anticoagulants
		platelet inhibitors
Main risk factors	alcohol/drugs consumption	
	low socio-economic status	
Main traumatic mechanism	Road traffic accidents	Falls <2 m
Risk of additional trauma	Multiple Major Trauma	Minor additional trauma
Main place of accident	Outdoor	Indoor
**Outcome**		
Death rate at 14 days in %	24.0%	40.9%
Disability at 1 year (median GOSE) *	7	7
Health-related quality of life at 1 year *		
Median physical component of SF-12	52.0	44.2
Median mental component of SF-13	51.4	52.3

GCS: Glasgow Coma Scale, ED: Emergency Department; HAIS: Head Abbreviated Injury Score; GOSE: Glasgow Outcome Scale extended; SF-12: General assessment instrument of health-related of quality of life. * Reference [30].

**Table 2 medsci-07-00012-t002:** Main missions and interventions of prehospital TBI care.

Aims	Issues	Diagnostic and Therapeutic Interventions
Assessment of TBI	Diagnosis based on probability	Scene observation, information from by-standers
	Neurological evaluation	GCS, pupil reaction, neurological signs of asymmetry, swallowing reflex
	Vital signs	HR, BP, SpO_2_, RR, signs of upper airway obstruction
Triage	Identify patients needing a specialized trauma center	Moderate or severe TBI
	Patient transport decision: Type of out-of-hospital EMS	Shortest delay to trauma center for moderate or severe TBI
	Identify dynamic changes in TBI severity	Repeated vital signs and neurological evaluations at regular intervals
Avoidance of secondary brain lesions	Avoid hypothermia	Maintain T > 35°C
	Avoid arterial hypotension (often related to extracranial hemorrhage)	SBP >110 mmHg, Fluid resuscitation: Isotonic solutions; not albumin
	Avoid hypoxemia	Maintain spO_2_ between 92 % and 95%, Consider prehospital intubation and normoventilation in patients with coma (GCS < 8), and altered swallowing reflex or hypoventilation

EMS: Emergency medical systems, HR: Heart Rate, BP: Blood Pressure, RR: Respiratory Rate, T: Remperature, SBP: Systolic blood pressure.

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
