# Peer review of "Prehospital and Emergency Care in Adult Patients with Acute Traumatic Brain Injury"

_medsci, 2019, doi:10.3390/medsci7010012_

Round 1

Reviewer 1 Report

This is a well written article. I do not have any major concern about the content of this manuscript. However, could the authors provide a graph or a table summarizing all the possible pre-hospital conditions based on the age? This could help the reading of this paper. 

Author Response

1.     This is a well written article. I do not have any major concern about the content of this manuscript.

Answer: Thank you!

2.     However, could the authors provide a graph or a table summarizing all the possible pre-hospital conditions based on the age? This could help the reading of this paper.

Answer:  We added a table 1 describing the difference between elderly and non-elderly adult patients in the early period after TBI. We included also relevant outcome data (see page 3, line 100)

Reviewer 2 Report

This is a potentially interesting paper but it lacks several important things that, if added would greatly enhance it.

First, there is no statement of purpose or scope of the paper. This leaves the reader wondering what the paper will entail and contributes to a lack of clarity

Likewise adding aims would be helpful

There is no mention of the inclusion/exclusion criteria for papers reviewed

Substantial editing for grammar and readability is suggested. 

Several sections (e.g. Diagnosis of TBI; Specificities of Pre-hospital and Emergency Care) have little to no references

Adding a figure and/or table that summarizing the main points would be helpful

Author Response

This is a potentially interesting paper but it lacks several important things that, if added would greatly enhance it.

1.     First, there is no statement of purpose or scope of the paper. This leaves the reader wondering what the paper will entail and contributes to a lack of clarity.

Answer: We added at the end of the introduction the aim of this narrative review. See page 2, line 48-52.

2.     Likewise adding aims would be helpful.

Answer: see point 1

3.     There is no mention of the inclusion/exclusion criteria for papers reviewed.

Answer:  In contrast to a systematic review a narrative review has normally no inclusion or exclusion criteria. With addition of the aim for this narrative review the chosen scientific literature is defined.

4.     Substantial editing for grammar and readability is suggested. 

Answer: The second authors is an English native speaker and a physician. He has checked again for any language errors.

5.     Several sections (e.g. Diagnosis of TBI; Specificities of Pre-hospital and Emergency Care) have little to no references

Answer:  We added 3 references in the section “Diagnosis of TBI”. We added 2 references in the section “Specificities of pre-hospital and emergency Care”

6.     Adding a figure and/or table that summarizing the main points would be helpful

Answer: We added a table 2 describing the main missions and interventions of prehospital TBI care (see page 6 line 248 and line 277).

Reviewer 3 Report

Thank you for your time and efforts reviewing the literature adult TBI in the prehospital and acute care setting. 

I think it would be useful to review and site the new guidelines for the management of TBI, 4th edition (Nancy Carney, Annette M. Totten, Cindy O'Reilly, Jamie S. Ullman, Gregory W.J. Hawryluk, Michael J. Bell, Susan L. Bratton, Randall Chesnut, Odette A. Harris, Niranjan Kissoon, Andres M. Rubiano, Lori Shutter, Robert C. Tasker, Monica S. Vavilala, Jack Wilberger, David W. Wright, Jamshid Ghajar;  Guidelines for the Management of Severe Traumatic Brain Injury, Fourth Edition, Neurosurgery, Volume 80, Issue 1, 1 January 2017, Pages 6–15, https://doi.org/10.1227/NEU.0000000000001432) used by physicians in the USA.  I would be very interested to see how your practice patterns and guidelines differ from ours.  It would also be helpful if you could formulate a table for the level of evidence your guidelines suggest similar to the tables in the above reference.

It would be helpful to add to the topic in line 322.  What procedure was used... ventricular drainage with ICP monitoring?  How are the outcomes different in monitored vs no monitored patients as well as ventricular drainage vs monitor only.

Line 327 and 328 can may be made more clear.  "Extradural" hematomas are epidural so it is hard to understand the meaning of this sentence.   

Author Response

Thank you for your time and efforts reviewing the literature adult TBI in the prehospital and acute care setting. 

1.     I think it would be useful to review and site the new guidelines for the management of TBI, 4th edition (Nancy Carney, Annette M. Totten, Cindy O'Reilly, Jamie S. Ullman, Gregory W.J. Hawryluk, Michael J. Bell, Susan L. Bratton, Randall Chesnut, Odette A. Harris, Niranjan Kissoon, Andres M. Rubiano, Lori Shutter, Robert C. Tasker, Monica S. Vavilala, Jack Wilberger, David W. Wright, Jamshid Ghajar;  Guidelines for the Management of Severe Traumatic Brain Injury, Fourth Edition, Neurosurgery, Volume 80, Issue 1, 1 January 2017, Pages 6–15, https://doi.org/10.1227/NEU.0000000000001432) used by physicians in the USA. 

Answer: The cited guidelines is focused on in-hospital TBI care. Our narrative review is focused on prehospital TBI are. See also reviewer 2 point 1 and changes on page 2 line 48-52.

2.     I would be very interested to see how your practice patterns and guidelines differ from ours. 

Answer: see point 1. For prehospital TBI care, the main difference is the staffing of out-of-hospital EMS which is mentioned in the section 3.1.2.

3.     It would also be helpful if you could formulate a table for the level of evidence your guidelines suggest similar to the tables in the above reference.

Answer: This narrative review of experts on prehospital TBI has not the intention to be a guideline on in-hospital care of TBI.

4.     It would be helpful to add to the topic in line 322.  What procedure was used... ventricular drainage with ICP monitoring?  

Answer: Based on the aim of this manuscript (see point 1) we deleted the parts on in-hospital care (section 3.6.3 and 3.6.4) and the main title of this section is now “Immediate therapeutic interventions” (see page 6 line 275).

5.     How are the outcomes different in monitored vs no monitored patients as well as ventricular drainage vs monitor only.

Answer: see point 4

6.     Line 327 and 328 can may be made more clear.  "Extradural" hematomas are epidural so it is hard to understand the meaning of this sentence.   

Answer: see point 4

Reviewer 4 Report

The authors report a review of prehospital and emergency care for patients with acute traumatic brain injury. Limited new information is provided. A clear objective for the manuscipt is not well defined. Organization of the paper is fair, at best. No significant new information is offered, and not new insights are gained from the material presented.

Author Response

The authors report a review of prehospital and emergency care for patients with acute traumatic brain injury. Limited new information is provided.

1.     A clear objective for the manuscript is not well defined.

Answer: See reviewer 2 point 1 and changes on page 2 line 48-52.

2.     Organization of the paper is fair, at best.

Answer: The structure of this narrative review corresponds to similar articles in this journal. See J Clin Med 2018 7:407 or Brain Sci 2018 8:170.

3.     No significant new information is offered, and not new insights are gained from the material presented.

Answer: The authors were invited to write a narrative review and not to present a study with new data on prehospital TBI care. We don’t agree with reviewer 4; for instance, this narrative review summarizes recent data on prehospital care of older patients with TBI which were very rarely presented. Furthermore, this narrative review summarizes recent models on early prediction after TBI. 

Round 2

Reviewer 2 Report

Much improved! Thank you for addressing the revision points thoroughly!

Reviewer 4 Report

The authors have revised their manuscript summarizing means to achieving more optimal pre-hospital care of adult patients with traumatic brain injury. They have addressed the suggestions made by the reviewers. The major changes that they have made since the initial submission include a clear objective and two tables, which summarize the major points that the authors make in their manuscript. A number of smaller changes have also been made. These changes have significantly improved the manuscript.